# OpenReview forum: "Dynamic Role-Graph Reinforcement Learning for Multi-Agent Collaborative Coding Systems"
_ICLR.cc/2026/Conference — Submitted to ICLR 2026_

### Official Review · Reviewer_7V75 · 2025-10-31

**Soundness:** 2
**Presentation:** 3
**Contribution:** 2
**Rating:** 2
**Confidence:** 3

**Summary:**

This paper introduces a MARL framework to address the challenges of coordination and role adaptation in collaborative software development teams. The authors propose a dynamic graph neural network architecture combined with role-aware reinforcement learning to model evolving team structures and optimize coding workflows. The framework integrates centralized critics with decentralized actors, role-conditioned attention mechanisms, and a collaboration complexity metric to guide policy adaptation. It also supports incremental updates to the graph structure, enabling smooth handling of agent churn. Experiments conducted on synthetic and real-world datasets demonstrate that DRGRL significantly outperforms traditional MARL methods.

**Strengths:**

The idea of using dynamic graph modeling with MARL, tailored specifically for collaborative coding environments, is insteresting. The role-graph duality, where roles are both learned and emergent, adds flexibility and realism to the representation of developer behaviors.

The framework’s ability to adapt to changing team compositions is useful.

The experimental results are robust, showing clear improvements over baselines, and the ablation studies provide convincing evidence of the contribution of each component.

**Weaknesses:**

The main concern is the weak innovation. This is a good practice of using MARL for collaboarative coding system but without much novel technqiue contributions.

The dynmaic graph is interesting and appropricate for the targeted problem but it is unclear how the MARL adapts to such structure change. Normally, the number of agents is fixed in MARL. The collaboration structure might change sometime but the agents are not changed.

**Questions:**

How does DRGRL perform in environments with highly heterogeneous agents or tasks that require frequent role switching?

How does DRGRL handle the "agents leaving and joining" in MARL?

---

### Official Review · Reviewer_QPve · 2025-10-31

**Soundness:** 1
**Presentation:** 1
**Contribution:** 2
**Rating:** 2
**Confidence:** 4

**Summary:**

The paper proposes a CTDE multi-agent RL framework for collaborative coding on a dynamic interaction graph. Key components include role-conditioned attention, a learned-plus-graph-derived "role duality" representation, and a global complexity scalar that interpolates exploration and exploitation. The system is evaluated on cited datasets with metrics such as TCR, MCF, CQS, and TEI. The experiments include component ablations and a churn-rate study.

**Strengths:**

1. Timely problem space (coordination under team churn).

2. Reasonable high-level scaffold (graph encoder $\rightarrow$ centralized critic; decentralized actors).

3. Includes some ablations and a churn-rate analysis.

**Weaknesses:**

1. The Novelty is limited. Role-conditioned attention is merely a minor adjustment to the scoring mechanism. "Role duality" simply involves concatenating learned features with graph statistics. The complexity gate functions as a hand-engineered temperature parameter. The work lacks supporting theoretical analysis, and the ablation studies fail to demonstrate the necessity of these proposed components.

2. The paper doesn't provide a precise formulation for Dec-POMDP and CTDE. The distinction between observations and critic inputs remains unclear, which gives rise to two issues, the risk of information leakage and unfair comparisons with baseline models.

3. The metrics TCR, MCF, CQS, and TEI are only defined verbally, with no accompanying mathematical equations to formalize their computation logic. The paper fails to specify the measurement units for these metrics, provide relevant references to support their design or prior use, and release the code for metric calculation. This lack of essential technical details makes the proposed metrics entirely non-auditable.

4. There are some issues in the experiments. The datasets are merely referenced in the paper, with their coverage scope and splitting methods left unspecified. The baselines are either outdated or misaligned with the research objectives, and the paper lacks both high-performance graph-Transformer multi-agent reinforcement learning (graph-Transformer MARL) models and tuned non-reinforcement learning (non-RL) schedulers. The ablation studies are inadequately designed, as they do not include confidence interval (CIs) analysis with multiple random seeds or sensitivity validation. The churn study lacks sufficient depth and fails to fully explore core influencing factors. The qualitative analysis in Section 5.5 only stays at the level of phenomenon description, without forming conclusions with diagnostic value.

5. The writing needs to be improved. Figures 1–2 are overly simplistic in content. References to these figures within the text are not hyperlinkable, as the "ref" command was not used. Proper punctuation is missing from the equations. Section 6 reads like filler content or boilerplate text generated by large language models, and the overall polish of the manuscript falls short of the standards expected for the ICLP venue.

**Questions:**

1. Can you provide a precise Dec-POMDP/CTDE specification: exact observation tensors at execution, critic inputs at training, the action vocabulary, reward decomposition, and the churn dynamics model (rates, sampling, and how churn events affect observations/actions/rewards)?

2. Can you give formal metric definitions with formulas, units, aggregation windows, and cite standards (or release metric code) so results are auditable?

3. Will you include modern baselines and report multi‑seed means and standard deviations (or confidence intervals) for all ablations?

4. Can you improve the representation by providing concrete diagrams that show design trade‑offs, clickable cross‑references, consistent notation, and a full appendix with hyperparameter tables, seeds, training curves, and dataset lists/splits?

---

### Official Review · Reviewer_5Lez · 2025-10-31

**Soundness:** 2
**Presentation:** 2
**Contribution:** 3
**Rating:** 2
**Confidence:** 4

**Summary:**

The paper proposes Dynamic Role-Graph Reinforcement Learning (DRGRL), a framework combining dynamic graph neural networks with role-aware reinforcement learning to model team coordination in collaborative coding. Agents (developers) form a dynamic graph where both structure and roles evolve. The model integrates role embeddings, a collaboration-complexity metric, and a centralized critic with decentralized actors.

**Strengths:**

1. Novel formulation: Integrates dynamic GNNs and MARL under a role-graph perspective; conceptually interesting.
2. Potential generalizability: The proposed abstraction could extend to other collaborative domains beyond coding.

**Weaknesses:**

1. Unclear role–coding linkage: The paper does not convincingly justify why “role modeling” is essential for collaborative coding. Concrete examples of how learned roles influence coding behavior are missing.

2. Ambiguous exposition: Several terms (e.g., “SK severing,” “collaboration complexity divider”) are undefined. Equations and symbols are not consistently explained.

3. Experimental simplicity: Evaluation relies on small-scale synthetic and limited open-source datasets. Missing comparisons with stronger baselines (e.g., recent role-based MARL or LLM-assisted collaboration frameworks). Some necessary ablation studies are needed to showcase the strengthes.

**Questions:**

1. How are “roles” inferred in practice, and how do they translate to concrete coding actions or IDE behaviors? Could you add some examples for clarification?

2. What is the computational cost of the dynamic attention mechanism on large teams?

3. How sensitive are results to team churn rates or graph density? Could any ablation studies or case studies better illustrate emergent role behaviors?

---

### Official Review · Reviewer_daBm · 2025-11-04

**Soundness:** 2
**Presentation:** 3
**Contribution:** 2
**Rating:** 2
**Confidence:** 3

**Summary:**

This paper addresses the limitations of static agentic workflows, which fail to adapt to the dynamic demands of real-world software development. To overcome this, the authors introduce a dynamic graph neural network (GNN) with a role-aware attention mechanism that adaptively updates agent interactions. In addition, they propose a novel role–graph duality module capable of learning roles directly from data or allowing them to emerge through graph dynamics. Experiments demonstrate the proposed method’s effectiveness in improving teamwork efficiency and code quality compared to baseline MARL approaches.

**Strengths:**

1. The paper is clearly written and easy to follow. The collaborative coding problem is well formulated, with clearly defined objectives and optimization targets. The proposed DRGRL framework is presented in a structured and understandable manner.
2. The paper includes both SOTA comparisons and ablation studies that support the authors’ claims.

**Weaknesses:**

1. **Limited novelty.** The GNN-based learning framework has been previously explored in works such as GPT-Swarm [1] and Aflow [2]. It remains unclear what the key distinctions are between DRGRL and these prior approaches. A more comprehensive comparison would help clarify the novel contributions.

2. **Insufficient experimental results.** The paper lacks comparisons with the latest SOTA methods on more public coding benchmarks. Including such comparisons would strengthen the empirical claims. In addition, the ablation studies are limited, analyzing the effects of agent count, edge density, iteration steps, and reward design would better illustrate the method’s behavior and robustness.

3. **Lack of qualitative analysis.** The paper would benefit from visual or case-based studies showing how agent interactions evolve over time, why these dynamics yield performance gains, and how roles emerge or adapt during training. Examples of roles learned from data versus those that arise from graph dynamics would provide valuable insight into the model’s interpretability.

4. **Missing model details.** Key implementation aspects are not fully described. For example, how is the role pool initialized? What does the inference process look like? Can the graph structure evolve during test time?

[1] Zhuge, M., Wang, W., Kirsch, L., Faccio, F., Khizbullin, D., & Schmidhuber, J. (2024, July). Gptswarm: Language agents as optimizable graphs. NeurIPS 2024.

[2] Zhang, J., Xiang, J., Yu, Z., Teng, F., Chen, X., Chen, J., ... & Wu, C. (2024). Aflow: Automating agentic workflow generation. ICLR 2025.

**Questions:**

1. The implementation details are unclear. How large is the collaboration graph? How do the number of agents and edges affect performance?

---

### Meta-Review · Area_Chair_rKB7 · 2026-01-04

**Summary:**

This paper proposes Dynamic Role-Graph Reinforcement Learning (DRGRL), a framework integrating dynamic graph neural networks with role-aware multi-agent reinforcement learning to address coordination in collaborative software development. It models developers as nodes in a dynamically evolving interaction graph, utilizing a centralized critic and decentralized actors to optimize team-level coding outcomes such as reduced merge conflicts and improved test coverage.

However, multiple reviewers identified fundamental flaws. Reviewers daBm, QPve, and 7V75 argued that the paper's novelty is significantly limited, with key components resembling minor adjustments to existing techniques rather than constituting a substantial methodological advance. Reviewers daBm, 5Lez, and QPve noted that the empirical validation is insufficient, citing a lack of comparisons with strong baselines, poorly defined evaluation metrics, reliance on small-scale datasets, and inadequate ablation studies. Reviewers 5Lez and QPve highlighted serious issues with clarity and presentation, including undefined terminology, missing mathematical formalisms, and overall writing quality below the expected standard for the venue.

Based on the significant and consistent concerns raised regarding novelty, experimental rigor, and presentation, I recommend a reject decision.

**Reviewer Concerns:**

This paper did not undergo a rebuttal process.

**Reviewer Scores:**

Since the authors did not submit a rebuttal, the scores should remain unchanged.

---

### Decision · Program_Chairs · 2026-01-26

Reject